# VORTEX: Physics-Driven Data Augmentations Using Consistency Training for Robust Accelerated MRI Reconstruction

**Arjun D Desai**[1][*]                                    ARJUNDD@STANFORD.EDU
**Beliz Gunel**[1][*]                                      BGUNEL@STANFORD.EDU
**Batu M Ozturkler**[1]                                    OZT@STANFORD.EDU
**Harris Beg**[1][†]                                       HARRIS@CALTECH.EDU
**Shreyas Vasanawala**[1]                          VASANAWALA@STANFORD.EDU
**Brian A Hargreaves**[1]                                  BAH@STANFORD.EDU
**Christopher Ré**[1]                                CHRISMRE@STANFORD.EDU
**John M Pauly**[1]                                        PAULY@STANFORD.EDU
**Akshay S Chaudhari**[1]                            AKSHAYSC@STANFORD.EDU
[1] *Stanford University*

**Editors:** Under Review for MIDL 2022

## Abstract

Deep neural networks have enabled improved image quality and fast inference times for various inverse problems, including accelerated magnetic resonance imaging (MRI) reconstruction. However, such models require a large number of fully-sampled ground truth datasets, which are difficult to curate, and are sensitive to distribution drifts. In this work, we propose applying *physics-driven* data augmentations for consistency training that leverage our domain knowledge of the forward MRI data acquisition process and MRI physics to achieve improved label efficiency and robustness to clinically-relevant distribution drifts. Our approach, termed VORTEX, (1) demonstrates strong improvements over supervised baselines with and without data augmentation in robustness to signal-to-noise ratio change and motion corruption in data-limited regimes; (2) considerably outperforms state-of-the-art purely image-based data augmentation techniques and self-supervised reconstruction methods on both in-distribution and out-of-distribution data; and (3) enables composing heterogeneous image-based and physics-driven data augmentations.

**Keywords:** MRI reconstruction, inverse problems, distribution shift, robustness, label-efficiency

## 1. Introduction

Magnetic resonance imaging (MRI) is a powerful diagnostic imaging technique; however, acquiring clinical MRI data typically requires long scan durations (30+ minutes). To reduce these durations, MRI data acquisition can be accelerated by undersampling the requisite spatial frequency measurements, referred to as *k-space* data. Reconstructing images without aliasing artifacts from such undersampled k-space measurements is ill-posed in the Hadamard sense (Hadamard, 1902). To address this challenge, previous methods utilized underlying image priors to constrain the optimization, most notably by enforcing sparsity in a transformation domain, in a process called compressed sensing (CS) (Lustig et al., 2008). However, CS methods provide limited acceleration and require long reconstruction times and parameter-specific tuning (Lustig et al., 2007; Akasaka et al., 2016).

Deep learning (DL) based accelerated MRI reconstruction methods have recently enabled higher acceleration factors than traditional methods, with fast reconstruction times and improved image

---

[*] Contributed equally
[†] Work done as Summer Undergraduate Research Fellow

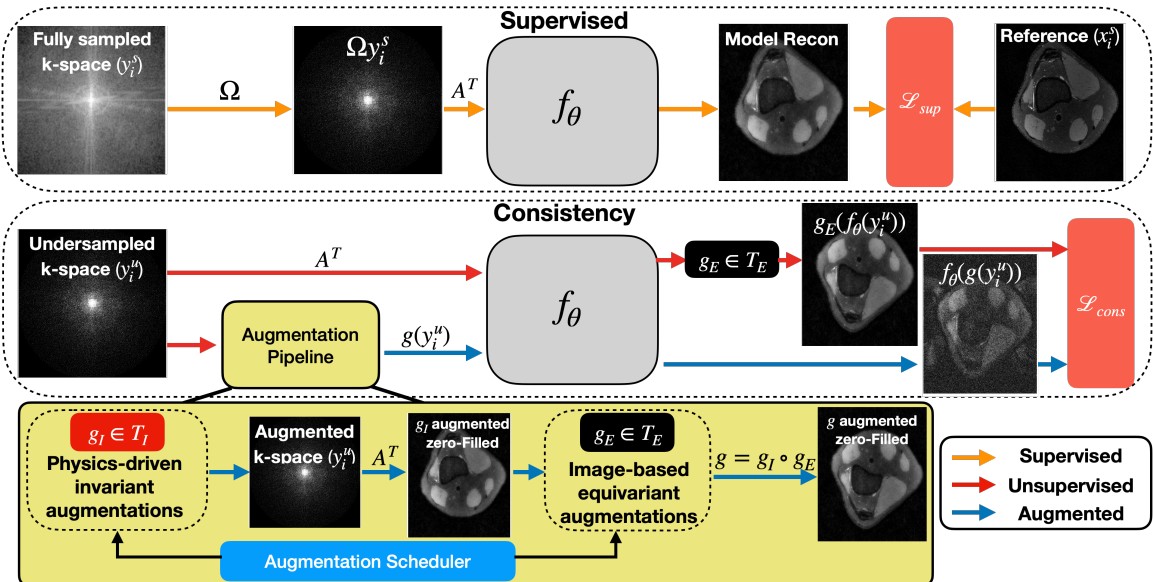

Figure 1: VORTEX, a semi-supervised consistency training framework for robust accelerated MRI reconstruction, enforces invariance to physics-driven and equivariance to image-based data augmentations, and supports curriculum learning and composing augmentations.

quality (Hammernik et al., 2018). However, these approaches rely on large amounts of paired undersampled and fully-sampled reference data for training, which is often costly or simply impossible to acquire. State-of-the-art reconstruction methods use large fully-sampled (supervised) datasets, with only a handful of methods leveraging prospectively undersampled (unsupervised) data (Chaudhari et al., 2020) or using image-based data augmentation schemes (Fabian et al., 2021) to mitigate data paucity. These DL-based MR reconstruction methods are also highly sensitive to clinically-relevant distribution drifts, such as scanner-induced drifts, patient-induced artifacts, and forward model changes (Darestani et al., 2021). Despite being a critical need, sensitivity to distribution drifts remains largely unexplored, with only a few studies that have studied simple forward model alterations such as undersampling mask changes at inference time (Gilton et al., 2021).

In this work, we leverage domain knowledge of the forward MRI data acquisition model and MRI physics through *physics-driven, acquisition-based* data augmentations for consistency training to build label-efficient networks that are robust to clinically-relevant distribution drifts such as signal-to-noise ratio (SNR) changes and motion artifacts. Our proposal builds on the Noise2Recon framework that conducts joint reconstruction for supervised scans and denoising for unsupervised scans (Desai et al., 2021a) by extending the original consistency denoising objective to a generalized data augmentation pipeline. Specifically, we propose a semi-supervised consistency training framework, termed VORTEX, that uses a data augmentation pipeline to enforce invariance to physics-driven data augmentations of noise and motion, and equivariance to image-based data augmentations of flipping, scaling, rotation, translation, and shearing (Fig. 1). VORTEX supports curriculum learning based on the difficulty of physics-driven augmentations and composing heterogeneous augmentations. We demonstrate this generalized consistency paradigm increases robustness to varying test-time perturbations without decreasing the reconstruction performance on non-perturbed, in-distribution data. We also show that VORTEX outperforms both state-of-the-art self-supervised training strategy SSDU (Yaman et al., 2020) and data augmentation scheme MRAugment (Fabian et al., 2021), which solely relies on image-based data augmentations. Unlike MRAugment, which requires preservation of training data noise statistics, VORTEX can operate on a broader family of MRI physics-based augmentations without noise statistics constraints.

Our contributions are the following: (1) We propose VORTEX, a semi-supervised consistency training framework for accelerated MRI reconstruction that enables composing image-based data augmentations with *physics-driven* data augmentations. VORTEX leverages domain knowledge of MRI physics and the MRI data acquisition forward model to improve label-efficiency and robustness. (2) We demonstrate strong improvements over supervised and self-supervised baselines in robustness to clinically-relevant distribution drifts of scanner-induced SNR change and patient-induced motion artifacts. Notably, we obtain +0.106 structural similarity (SSIM) and +5.3dB complex PSNR (cPSNR) improvement over supervised baselines on heavily motion-corrupted scans in label-scarce regimes. (3) We outperform MRAugment, a state-of-the-art purely image-based data augmentation technique for MRI reconstruction. We achieve improvements of +0.061 SSIM and +0.2dB cPSNR on in-distribution data, +0.089 SSIM and +2.5dB cPSNR on noise-corrupted data, and +0.125 SSIM and +7.8dB cPSNR on motion-corrupted data. (4) We conduct ablations comparing image space and latent space consistency and designing curricula for data augmentation difficulty. Our code and experiments are publicly available[1].

## 2. Related Work

Supervised accelerated MRI reconstruction methods (Adler and Öktem, 2018; Aggarwal et al., 2018; Sandino et al., 2020) rely on a large corpus of fully-sampled scans. Although lagging in performance to supervised techniques, several studies have leveraged unsupervised data including using generative adversarial networks (Lei et al., 2020; Cole et al., 2020), self-supervised learning (Yaman et al., 2020), and dictionary learning (Lahiri et al., 2021). Fabian et al. (2021) proposed image-based data augmentations to reduce dependence on supervised training data. While Darestani et al. (2021) demonstrated that both trained and untrained (Darestani and Heckel, 2021) reconstruction methods exhibit sensitivity to adversarial perturbations and distribution drifts, no mitigation approaches were discussed. Consistency training was initially proposed to build noise invariance in input data (Miyato et al., 2018; Sajjadi et al., 2016; Clark et al., 2018) or hidden representations (Bachman et al., 2014; Laine and Aila, 2016). Desai et al. (2021a) extended these methods to a consistency training framework for joint MRI image reconstruction and denoising, where additive noise was applied to undersampled k-space. Beyond noise-based consistency, Xie et al. (2020) showed that semantics-preserving data augmentation consistency (RandAugment (Cubuk et al., 2020) for vision tasks and back-translation for language tasks (Edunov et al., 2018)) led to significant performance boosts. Pawar et al. (2019) proposed a supervised DL method to map simulated motion-corrupted scans to clean scans as a post-processing method after reconstruction. Liu et al. (2020) extended iterative application of image denoisers as imaging priors (Romano et al., 2017) for general artifact removal such as that of motion. Gan et al. (2021) extended this method by training the model in the measurement domain without supervised data. However, these methods require multiple measurements of the same object undergoing nonrigid deformations, which is clinically infeasible. An extended discussion on related work in comparison to our work is available in Appendix B.

## 3. Background on Accelerated Multi-coil MRI Reconstruction

We consider the clinically-relevant case of accelerated multi-coil MRI acquisition where multiple receiver coils are used to acquire spatially-localized undersampled k-space measurements modulated by corresponding sensitivity maps (Pruessmann et al., 1999). The undersampling operation can be

---

1. https://github.com/ad12/meddlr

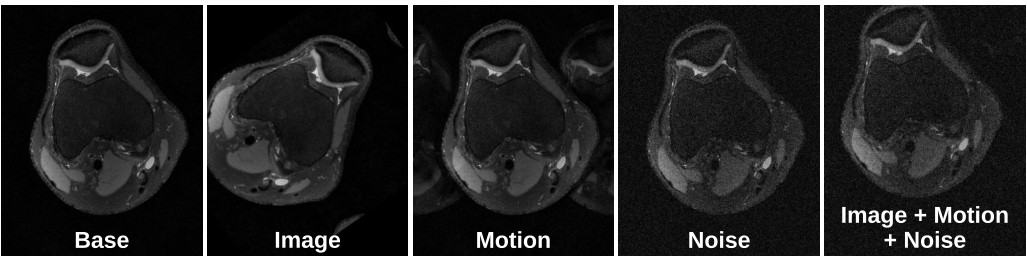

Figure 2: Sample image-based, physics-driven (motion where $\alpha = 0.2$, noise where $\sigma = 0.2$), and composed (image + physics) augmentations applied to a fully-sampled image.

represented by a binary mask $\Omega$ that indexes acquired samples in k-space. The forward problem for multi-coil accelerated MRI can be written as $y = \Omega \boldsymbol{F} \boldsymbol{S} x^* + \epsilon = Ax^* + \epsilon$, where $y$ is the measured signal in k-space, $\boldsymbol{F}$ is the discrete Fourier transform matrix, $\boldsymbol{S}$ is the receiver coil sensitivity maps, $x^*$ is the ground-truth signal in image-space, and $\epsilon$ is additive complex Gaussian noise. $A = \Omega \boldsymbol{F} \boldsymbol{S}$ is the known forward operator during acquisition (see Appendix A for notation). Because this problem is ill-posed (Hadamard, 1902), the underlying image $x^*$ cannot be recovered uniquely without regularization (e.g. sparsity in compressed sensing (Lustig et al., 2008)).

## 4. Methods

We propose VORTEX, a semi-supervised consistency training framework that integrates a generalized data augmentation pipeline for accelerated MRI reconstruction (Fig. 1). We consider the setup with dataset $\mathcal{D}$ that consists of (1) fully-sampled examples in k-space $y^{(s)}$ with corresponding supervised reference ground truth images $x^{(s)}$, and (2) undersampled-only k-space examples $y^{(u)}$. $f_\theta$ is the learned reconstruction model with the forward operator $A$. A pixel-wise $\ell_1$ supervised loss $\mathcal{L}_{sup}$ is computed for supervised examples $y^{(s)}$. Undersampled examples $y^{(u)}$ are passed through the *Augmentation Pipeline* (see §4.1). We consider the case where there are more unsupervised examples than supervised examples, a common observation in clinical practice. Let $T_I$ denote the set of invariant transformations consisting of physics-driven data augmentations such as additive complex Gaussian noise and motion corruption (see §4.1.1). Similarly, let $T_E$ denote the equivariant transformations that include image-based data augmentations such as flipping, rotation, translation, scaling, and shearing (see §4.1.2). We define our use of the terms *invariance* and *equivariance* and how these definitions motivate the structure of augmentations in Appendix C. A pixel-wise $\ell_1$ consistency loss $\mathcal{L}_{cons}$ is computed between the model reconstruction outputs of input undersampled examples with and without augmentation. The overall training objective is the following:

$$\mathcal{L}_{\text{VORTEX}} = \sum_i \|f_\theta(y_i^s, A) - x_i^s)\|_1 + \lambda \mathcal{L}_{cons}$$

$$\text{where} \quad \mathcal{L}_{cons} = \begin{cases} \|f_\theta(y_i^u, A) - f_\theta(g(y_i^u), A)\|_1, & \text{if } g \in T_I \\ \|g(f_\theta(y_i^u, A)) - f_\theta(g(y_i^u), A)\|_1, & \text{if } g \in T_E \end{cases}$$

### 4.1. Generalized Data Augmentation Pipeline for Consistency Training

Our augmentation pipeline enables composing state-of-the-art image-based data augmentations with physics-driven data augmentations motivated by the MRI data acquisition forward model.

#### 4.1.1. PHYSICS-DRIVEN DATA AUGMENTATIONS

**Noise.** As one of the most common MRI artifacts, practical MRI reconstruction methods should be robust to SNR variations. We leverage noise for consistency training since it can be well modeled in the MRI data acquisition forward model (Macovski, 1996). Specifically, we sample $\sigma$ from a range

$\mathcal{R}(\sigma) = [\sigma^{LN}, \sigma^{HN})$ where **LN** (light noise) and **HN** (heavy noise) are chosen based on visual inspections of clinical scans by a board-certified clinical radiologist. We normalize each sampled $\sigma$ to the magnitude of the image to induce the same relative SNR changes across scans. We denote the operation of adding noise to the k-space as $g_N$, where the noise-augmented unsupervised example is given by $g_N(y_i^{(u)}) = y_i^{(u)} + \epsilon_i$. We provide an example of a noise-augmented scan in Fig. 2.

**Motion.** Rigid patient motion during MRI scans degrades image quality and causes ghosting artifacts, which considerably affects the quality of images in pediatric or claustrophobic patients. While navigator-based sequences that sample low-resolution motion can be used for motion correction, they require custom sequences that often lead to increased acquisition time, reduced SNR, and complicated reconstruction (Zaitsev et al., 2001). Many multi-shot MRI acquisitions sample data over multiple *shots* where consecutive k-space lines are acquired in separate excitations (Anderson and Gore, 1994). Here, motion across every shot manifests as additional phase in k-space and as translation in image space. Thus, one-dimensional translational motion artifacts across the phase dimension can be modeled using random phase errors that alter odd and even lines of k-space separately. We leverage motion for consistency training since we can precisely model rigid motion in k-space. We denote the phase error due to motion for $i^{th}$ example by $e^{-j\phi_i}$ that corresponds to a translational motion. We sample two random numbers from the uniform distribution $m_o, m_e \sim U(-1, 1)$ which is chosen from a specified range $\mathcal{R}(\alpha) = [\alpha^{LM}, \alpha^{HM})$ where $\alpha$ denotes the amplitude of the phase errors and **LM** (light motion) and **HM** (heavy motion) are chosen based on visual inspections of clinical scans by a board-certified clinical radiologist. For a given k-space readout $k^{th}$, the phase error is:

$$\phi_i^k = \begin{cases} \pi\alpha m_o, & \text{if k is odd} \\ \pi\alpha m_e, & \text{if k is even} \end{cases}$$

We denote the operation of adding motion to the k-space as $g_M$, in which case the motion-augmented unsupervised example is given by $g_M(y_i^{(u)}) = y_i^{(u)} e^{-j\phi_i}$ (example scan shown in Fig. 2).

### 4.1.2. IMAGE-BASED DATA AUGMENTATIONS

In contrast to classification problems where labels are invariant with respect to the augmentations, data augmentations in the MR reconstruction task need to transform the target images and their corresponding k-space and coil sensitivity measurements. Unlike physics-driven augmentations that occur in k-space, image-based augmentations occur in the image domain. Since the training data initially exists as k-space measurements, we transform it into the image domain using coil sensitivity maps, and subsequently apply a cascade of the image-based data augmentations to both the image and the sensitivity maps. Image-based data augmentations include pixel-preserving augmentations such as flipping, translation, arbitrary and 90 degree multiple rotations, translation, as well as isotropic and anisotropic scaling. Using the augmented image and transformed sensitivity maps, we run the forward operator $A$ to generate the corresponding undersampled k-space measurements.

**Composing Augmentations.** Our *Augmentation Pipeline* allows for composing different combinations of physics-driven and image-based data augmentations, with example composed augmentations shown in Fig. 2. It is important to note that composing multiple physics-driven augmentations such as noise and motion corruption represents a real-world scenario as multiple artifacts can occur simultaneously during MRI acquisition. Appendix C discusses augmentation composition in detail.

## 4.2. Augmentation Scheduling

We adopt curriculum learning (Hacohen and Weinshall, 2019) for physics-driven data augmentations, where we seek to schedule the *task difficulty*. Difficulty is denoted by $\sigma$, the standard deviation of

Table 1: Average test results for in-distribution data and out-of-distribution data with heavy motion ($\alpha$=0.4) and heavy noise ($\sigma$=0.4) perturbations. Physics augmentations are compositions of noise and motion in their *heavy* training difficulty configurations.

| Perturbation | None | | Motion (heavy) | | Noise (heavy) | |
|---|---|---|---|---|---|---|
| Model | SSIM | cPSNR (dB) | SSIM | cPSNR (dB) | SSIM | cPSNR (dB) |
| Compressed Sensing | 0.847 (0.011) | 35.4 (0.4) | 0.724 (0.090) | 24.5 (1.4) | 0.708 (0.014) | 30.5 (0.2) |
| Supervised | 0.798 (0.038) | 35.8 (0.4) | 0.706 (0.048) | 27.0 (0.8) | 0.807 (0.015) | 32.2 (0.3) |
| MRAugment | 0.811 (0.043) | 36.2 (0.5) | 0.660 (0.040) | 24.0 (1.0) | 0.742 (0.005) | 30.8 (0.3) |
| SSDU | 0.787 (0.026) | 34.9 (0.4) | 0.734 (0.009) | 31.9 (1.7) | 0.716 (0.023) | 32.5 (0.3) |
| Aug (Physics) | 0.789 (0.045) | 35.7 (0.4) | 0.739 (0.010) | 31.9 (2.4) | 0.739 (0.051) | 33.4 (0.3) |
| Aug (Image+Physics) | 0.785 (0.050) | 36.1 (0.5) | 0.742 (0.022) | **32.8 (2.4)** | 0.727 (0.051) | 33.7 (0.4) |
| VORTEX (Image) | 0.862 (0.030) | 36.4 (0.3) | 0.648 (0.080) | 26.1 (0.7) | 0.767 (0.016) | 31.5 (0.2) |
| VORTEX (Physics) | **0.872 (0.033)** | **36.4 (0.3)** | **0.785 (0.019)** | 31.8 (2.8) | 0.817 (0.034) | **33.9 (0.2)** |
| VORTEX (Image+Physics) | 0.861 (0.036) | 36.4 (0.4) | 0.777 (0.034) | 31.1 (2.7) | **0.831 (0.023)** | 33.3 (0.1) |

the additive zero-mean complex-valued Gaussian noise, and $\alpha$, the amplitude of the phase errors for motion. Note that this is in contrast to the MRAugment scheduling strategy, which only schedules the probability $p$ of an augmentation. Concretely, for noise, we consider a time-varying range $\mathcal{R}(\sigma(t)) = [\sigma^{L}, \sigma^{H}(t))$, where $t$ indexes the iteration number during training. The upper-bound $\sigma^{H}(t)$ increases monotonically to ensure task difficulty increases during training. We consider two scheduling techniques $\beta(t)$ such that $\sigma^{H}(t) = \sigma^{L} + \beta(t)(\sigma^{H} - \sigma^{L})$: (1) **Linear:** $\beta(t) = t/M$, and (2) **Exponential:** $\beta(t) = \frac{1-e^{-t/\tau}}{1-e^{-M/\tau}}$, where $M$ is the number of epochs until which task difficulty increases and $\tau$ is the time-constant for exponential scheduling. After $M$ epochs, training proceeds with constant upper bound $\sigma^{H}$. Scheduling for motion is the same where $\sigma$ is replaced with $\alpha$, and image-based data augmentations follow the scheduling strategy proposed in MRAugment as there is no explicit sense of difficulty for that family of data augmentations. Fig. 5 in Appendix E.2 shows simulated $\beta(t)$ for different curricula configurations.

## 5. Experiments

We evaluate our method using the publicly available mridata 3D fast-spin-echo (FSE) multi-coil knee dataset (Ong et al., 2018). 3D MRI scans were decoded into a hybrid k-space ($x \times k_y \times k_z$) using the 1D orthogonal inverse Fourier transform along the readout direction $x$. All methods reconstructed 2D $k_y \times k_z$ slices. Sensitivity maps were estimated for each slice using JSENSE (Ying and Sheng, 2007). 2D Poission Disc undersampling masks were used for training and evaluation. $N_s$ training scans were randomly selected to be fully-sampled (supervised) examples while $N_u$ scans were used to simulate undersampled-only scans. All methods used 2D U-Net network with a complex-$\ell_1$ training objective for both supervised and consistency losses. Appendix D discusses the experimental setup in further detail, and Appendix F includes additional experiments across all methods on the 2D fastMRI multi-coil brain dataset (Zbontar et al., 2018). Appendix E details ablation experiments comparing latent space vs pixel-level consistency and variations in augmentation scheduling.

### 5.1. Robustness to Clinically Relevant Distribution Drifts

Unlike many other ML domains, the source of possible distribution drifts in accelerated MRI can be well characterized and simulated based on domain knowledge of MRI physics. In this work, we simulate SNR and motion corruptions, two common MRI artifacts, at inference time using models described in §4.1.1. Specifically, we use $\sigma = 0.2$ for light noise, $\sigma = 0.4$ for heavy noise, $\alpha = 0.2$ for light motion, and $\alpha = 0.4$ for heavy motion. In Table 1, we compare VORTEX's performance for in-distribution and OOD data at 16x acceleration to supervised methods using both physics-driven and the state-of-the-art image-based MRAugment augmentations, and to the

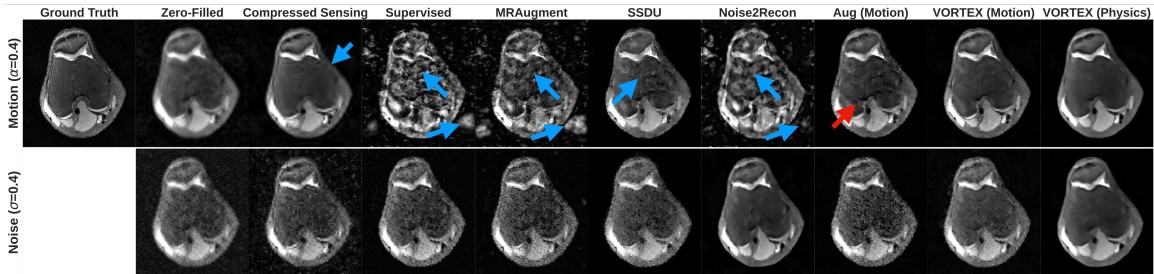

Figure 3: Example reconstructions for simulated scans with heavy motion (top) and heavy noise (bottom). Compressed sensing (CS), *Supervised*, MRAugment, SSDU, and Noise2Recon amplify motion ghosting artifacts (blue arrow). While supervised training with motion augmentations (*Aug (Motion)*) reduces these artifacts, it still suffers from artifacts (red arrow) and extensive blurring. *VORTEX (Motion)* and *VORTEX (Physics)* (i.e. Motion+Noise) suppress these artifacts. Methods without noise augmentations (CS, *Supervised*, MRAugment, SSDU, Aug (Motion), VORTEX (Motion)) amplify image noise. *VORTEX (Physics)* suppresses noise without over-blurring the image.

state-of-the-art self-supervised via data undersampling (SSDU) reconstruction method (Yaman et al., 2020). We describe all baseline implementations in detail in Appendix D.2. We isolate the benefits of consistency training with VORTEX from the utility of the augmentations *(Aug)* themselves by separately comparing MRAugment (i.e. *Aug (Image)*), *Aug (Physics)*, and *Aug (Image + Physics)* (details in Appendix D.3.1). In Table 2, we compare supervised baselines without and with the physics-based augmentations to VORTEX at different OOD motion and noise levels at inference time. We note that Noise2Recon is a specialized case of the VORTEX framework (i.e *VORTEX (Noise)*) where the Augmentation Pipeline only consists of noise augmentations. For the supervised training with augmentation methods, augmentation is applied with probability $p = 0.2$ during training for noise, motion, and composition corresponding to $g_N(g_M(\cdot))$. For enforcing consistency, we used $\lambda = 0.1$ for $\mathcal{L}_{cons}$ weighting for noise, motion, and composition that we refer to as *Physics*. Both *Aug (Motion)* and *VORTEX (Motion)* models were trained with $\mathcal{R}(\alpha) = [0.2, 0.5)$, and both *Aug (Noise)* and *VORTEX (Noise)* models were trained with $\mathcal{R}(\sigma) = [0.2, 0.5)$. *Aug (Physics)* and *VORTEX (Physics)* setting also follow these ranges. We used a balanced data sampling approach where unsupervised and supervised examples are sampled at a fixed ratio of 1:1 during training (Desai et al., 2021a). All consistency training approaches used augmentation curricula with highest validation cPSNR as described in §4.2. Results are shown with 5x more unsupervised slices than supervised (1600 vs 320), which is a realistic clinical scenario. We show results for different accelerations, training times and augmentation curricula in Appendices D and E.

## 5.2. VORTEX vs. Baseline Results

As shown in Table 1, *VORTEX (Physics)* demonstrated substantial improvements of +0.074 SSIM and +0.6dB cPSNR over the *Supervised* baseline, +0.061 SSIM and +0.2dB cPSNR over *MRAugment*, and +0.085 SSIM and +1.5dB cPSNR over *SSDU* for in-distribution data. As *VORTEX (Image)* also considerably improves over *Supervised* and *MRAugment*, a dominant mechanism of the benefits may be attributed to the consistency training even for the in-distribution setting. For both heavy motion and noise settings, including physics augmentations is vital for robust performance, as *MRAugment*, *SSDU*, and *VORTEX (Image)* perform worse, even compared to the *Supervised* baseline. For heavy motion, we show an improvement of +0.079 SSIM and +4.8dB cPSNR over the *Supervised* and +0.125 SSIM and +7.8dB cPSNR over *MRAugment* with *VORTEX (Physics)*. Similarly, for heavy noise, we show an improvement of +0.024 SSIM and +1.1dB cPSNR over the Supervised baseline

Table 2: Results at 16x acceleration for in-distribution and OOD with SNR and motion perturbations. VORTEX uses curriculum learning (see Appendix E.2).

| Perturbation | Metric | Supervised | Aug (Motion) | Aug (Noise) | Aug (Physics) | VORTEX (Motion) | Noise2Recon (i.e. VORTEX (Noise)) | VORTEX (Physics) |
|---|---|---|---|---|---|---|---|---|
| None | SSIM | 0.798 (0.038) | 0.793 (0.041) | 0.805 (0.045) | 0.789 (0.045) | 0.877 (0.029) | **0.882 (0.031)** | 0.869 (0.030) |
| | cPSNR (dB) | 35.8 (0.4) | 35.9 (0.5) | 35.8 (0.4) | 35.7 (0.4) | 36.4 (0.3) | **36.4 (0.3)** | 36.4 (0.3) |
| Motion (light) | SSIM | 0.809 (0.028) | 0.793 (0.028) | 0.799 (0.039) | 0.785 (0.036) | **0.867 (0.021)** | 0.854 (0.015) | 0.854 (0.029) |
| | cPSNR (dB) | 33.6 (0.2) | 35.1 (0.5) | 34.1 (0.4) | 35.0 (0.4) | **35.8 (0.5)** | 32.8 (1.3) | 35.4 (0.1) |
| Motion (heavy) | SSIM | 0.706 (0.048) | 0.751 (0.025) | 0.722 (0.042) | 0.739 (0.010) | **0.812 (0.021)** | 0.731 (0.014) | 0.803 (0.017) |
| | cPSNR (dB) | 27.0 (0.8) | 31.5 (2.7) | 29.6 (1.4) | 31.9 (2.4) | **32.3 (2.5)** | 27.1 (1.7) | 32.3 (2.6) |
| Noise (light) | SSIM | 0.830 (0.024) | 0.786 (0.032) | 0.778 (0.049) | 0.761 (0.049) | **0.857 (0.015)** | 0.854 (0.033) | 0.840 (0.034) |
| | cPSNR (dB) | 33.8 (0.3) | 33.7 (0.3) | 34.2 (0.3) | 34.2 (0.3) | 34.0 (0.1) | **34.8 (0.2)** | 34.8 (0.2) |
| Noise (heavy) | SSIM | 0.807 (0.015) | 0.758 (0.024) | 0.745 (0.054) | 0.739 (0.051) | 0.823 (0.008) | **0.830 (0.033)** | 0.812 (0.033) |
| | cPSNR (dB) | 32.2 (0.3) | 32.0 (0.3) | 33.5 (0.3) | 33.4 (0.3) | 32.4 (0.1) | **34.0 (0.3)** | 33.9 (0.3) |

and +0.089 SSIM and +2.5dB cPSNR over *MRAugment* with *VORTEX (Image + Physics)*. In Table 2, we observe consistent improvements over both *Supervised* and *Aug* baselines for light and heavy motion cases, with a large improvement of +0.106 SSIM and +5.3dB cPSNR with *VORTEX (Motion)* over *Supervised* for heavy motion-corruptions, demonstrating the strength of our method at varying levels of OOD corruptions at inference time. Also, we depict a large improvement of +0.079 SSIM/+0.6dB pSNR with *VORTEX (Motion)* and +0.084 SSIM/+0.6dB cPSNR with *VORTEX (Noise)* compared to the supervised baseline for in-distribution data while none of the *Aug* baselines show any meaningful improvement. Example reconstructions are shown in Fig. 3.

The substantial performance gain with VORTEX in both in-distribution and OOD settings suggests that the consistency training framework is amenable to both image-based and physics-driven augmentations. While supervised training requires noise statistics-preserving augmentations, consistency training can relax this constraint and allow for more diverse augmentations (see Appendix B.1). We highlight that our proposed consistency-based improvements are considerably larger than prior reported values for DL methods that use different architectures, loss functions, or data consistency schemes (Zbontar et al., 2018; Hammernik et al., 2021). We also note that SSIM is clinically preferred to cPSNR for quantifying perceptual quality (Knoll et al., 2020).

## 6. Conclusion

We propose VORTEX, a semi-supervised consistency training framework for accelerated MRI reconstruction that uses a generalized data augmentation pipeline for improved label-efficiency and robustness to clinically relevant distribution drifts. VORTEX enforces invariance to *physics-driven, acquisition-based* augmentations and enforces equivariance to image-based augmentations, enables composing data augmentations of different types, and allows for curriculum learning based on the difficulty of physics-driven augmentations. We demonstrate strong improvements over fully-supervised baselines and state-of-the-art data augmentation (MRAugment) and self-supervised (SSDU) methods on both in-distribution and OOD data. Our framework is model-agnostic and could be used with any other MRI reconstruction models or even for other image-to-image tasks with appropriate data augmentations. Besides the strengths of our method, we also note several limitations. While our framework is flexible to work with any motion model, we utilized a simpler 1D motion model. We also did not evaluate VORTEX on prospective data or consider complex artifacts that are challenging to model such as $B_0$ variations. In future work, we plan to extend physics-driven, acquisition-based augmentations to account for more complex motion models, additional OOD MRI artifacts, and non-Cartesian undersampling patterns in prospectively acquired clinical data to work towards building robust DL-based MR reconstruction models that can be safely deployed in clinics.

## Acknowledgments

This work was supported by R01 AR063643, R01 AR AR077604, R01 EB002524, R01 EB009690, R01 EB026136, K24 AR062068, and P41 EB015891 from the NIH; the Precision Health and Integrated Diagnostics Seed Grant from Stanford University; DOD – National Science and Engineering Graduate Fellowship (ARO); National Science Foundation (GRFP-DGE 1656518, CCF1763315, CCF1563078); Stanford Artificial Intelligence in Medicine and Imaging GCP grant; Stanford Human-Centered Artificial Intelligence GCP grant; GE Healthcare and Philips.

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

## Appendix A. Glossary

Table 3 provides the notation used in the paper.

Table 3: Summary of notation used in this work.

|  | Notation | Description |
|---|---|---|
| **MRI forward model** | $x, y$ | Image, k-space measurements |
|  | $y_i^{(s)}, y_i^{(u)}$ | Fully-sampled (supervised) k-space, prospectively undersampled (unsupervised) k-space |
|  | $\Omega, \boldsymbol{F}, \boldsymbol{S}$ | Undersampling mask, fourier transform matrix, coil sensitivity maps |
|  | $A$ | The forward MRI acquisition operator |
|  | $\epsilon$ | Additive complex-valued Gaussian noise |
| **Augmentation transforms** | $T$ | Set of data transforms |
|  | $T_I, T_E$ | Set of invariant and equivariant data transforms |
|  | $g, g_E, g_I$ | Transform, equivariant transform, invariant transform |
|  | $\bar{G}_E, \bar{G}_I$ | Sequence of sampled invariant and equivariant data transforms |
|  | $\mathcal{N}(0, \sigma)$ | Complex gaussian distribution with zero-mean, variance $\sigma^2$ |
|  | $\alpha$ | Motion-induced phase error amplitude |
|  | $\phi_i^k$ | Phase error for $k^{\text{th}}$ phase encode line in example $i$ |
|  | $\mathcal{R}(\cdot)$ | Range |
|  | $\beta(t)$ | Difficulty scale |
|  | LM, HM | Light motion ($\alpha$=0.2), heavy motion ($\alpha$=0.4) |
|  | LN, HN | Light noise ($\sigma$=0.2), heavy noise ($\sigma$=0.4) |
| **Model components and losses** | $\mathcal{L}_{sup}, \mathcal{L}_{cons}$ | Supervised, consistency loss |
|  | $\lambda$ | Consistency loss weight |
|  | $R_i$ | U-Net resolution level $i$ |

## Appendix B. Extended Related Work

In this section, we summarize the key differences between VORTEX and prior work in data augmentations (i.e. MRAugment) and in consistency training (i.e. Noise2Recon). Specifically, we highlight two advantages of VORTEX:

1. **Image-based *and* Acquisition-based Augmentations.** VORTEX can relax the assumption that augmentations must preserve the noise statistics of the data (Fabian et al., 2021). This allows VORTEX to leverage both image-based and acquisition-based augmentations, which do not preserve the noise statistics of the data.

2. **Regularization Beyond Noise.** VORTEX can leverage physics-driven augmentations beyond the standard denoising regularization used in prior work in both consistency (Desai et al., 2021a) and pre-training (Romano et al., 2017). Thus, it is feasible to extend VORTEX to other relevant clinical artifacts while maintaining the regularization properties of the well-studied denoising task.

### B.1. VORTEX vs MRAugment

MRAugment proposes a framework for applying image-based augmentations on fully-supervised training data. This approach showed improved performance in data-limited settings, which may suggest the family of image-based augmentations are helpful in reducing model overfitting. It also suggests scheduling the likelihood of applying an augmentation can be helpful for reducing the number of augmented examples in early stages of training.

**Image vs Acquisition Augmentations.** MRAugment focuses on the use of image-based augmentations for supervised training. In VORTEX, both image-based and MRI acquisition-based augmentations are used for semi-supervised consistency training to 1) reduce dependence on supervised training data and 2) increase robustness to physics-driven perturbations that are frequently observed during MRI acquisition.

**Relaxing Assumption of Preserved Noise Statistics.** MRAugment notes that the family of image-based augmentations were selected to ensure that noise statistics of the training data were preserved. However, this constraint excludes acquisition-based augmentations, particularly noise and motion, which are needed to build robustness to noise and motion artifacts in MRI. However, these acquisition-based augmentations inherently change the effective noise floor (and thus SNR) of the scan, and thus violate this constraint. We empirically validate this claim in supervised settings, where acquisition-based augmentations perform worse than standard supervised training in in-distribution settings (Table 2). This tradeoff between in-distribution performance and OOD robustness would preclude the application of acquisition-based augmentations in practice.

However, with VORTEX, not only is this tradeoff mitigated but the performance in both in-distribution and OOD settings is significantly improved (Table 10). This improved performance empirically demonstrates that the assumption that augmentations must preserve noise statistics can be relaxed in the VORTEX framework. Thus, both image-based and acquisition-based augmentations can be leveraged simultaneously, which leads to improvements in performance over either family of augmentations alone (Table 10).

**Precomputing Coil Sensitivity Maps.** Integrating coil sensitivity maps is standard clinical practice to help constrain the optimization problem for MRI image reconstruction (Sandino et al., 2021; Robson et al., 2008; Roemer et al., 1990). MRAugment utilizes the end-to-end VarNet, which *learns* to jointly estimate coil sensitivities and reconstruct images (Sriram et al., 2020). Thus, the augmentation pipeline in MRAugment does not need to explicitly account for the effect of image-based transformations on sensitivity maps. It also has the added benefit of optimzing sensitivity map estimtion with respect to augmented data. In practice, precomputing coil sensitivities is feasible and routine with sensitivity map estimation methods such as ESPIRiT (Uecker et al., 2014) and JSENSE (Ying and Sheng, 2007). Additionally, precomputed maps are important in multi-coil datasets where the number of coils are not constant across different scans, which is critical when patients with heterogenous anatomies are being imaged (Desai et al., 2021b).

VORTEX utilizes precomputed sensitivity maps estimated from auto-calibration regions in each scan. Because image-based augmentations are designed to emulate shifts in the imaging target, they also impact the coil geometry and sensitivity maps that are estimated. In contrast to the MRAugment sensitivity map formulation, which assumes sensitivity maps are fixed, VORTEX integrates physics-based modeling to appropriately warp sensitivity maps based on image-based augmentations. Given some equivariant image-based transform $g_E$, the augmented image for coil $i$ ($\tilde{x}_i$) can be defined as

$$\tilde{x}_i = g_E(\boldsymbol{S}_i)g_E(x)$$

**Scheduling Augmentation Difficulty.** MRAugment and VORTEX also differ in the mechanism of how augmentations are scheduled. MRAugment proposes an augmentation scheduling method that schedules the probability of applying an augmentation. Thus, training can occur predominantly on collected data in earlier stages of training and augmentations can help reduce overfitting at later training stages.

VORTEX is designed to build robustness to OOD perturbations, where the *extent* (and, more generally, *difficulty*) of these perturbations will be unknown at test time. In this framework, augmentations must not only function as a regularization method for improved performance on in-distribution data, but also appropriately model a separate distribution of data with respect to which the model can be trained. Thus, the model must learn to jointly optimize for both in-distribution (default training data) and OOD (perturbation-corrupt data) examples simultaneously. Intuitively, we need to design an augmentation scheduling scheme that will allow the model to gradually learn to generalize to higher extents (more difficult) perturbations over time while still ensuring examples from both distributions are sampled for joint optimization. To ensure that augmentations are always applied but at different extents, we propose a curriculum learning strategy for scheduling the *difficulty* of the augmentation.

### B.2. VORTEX vs Noise2Recon

Noise2Recon proposes a semi-supervised consistency based framework for joint denoising and reconstruction. This approach showed improved performance in label-limited settings, where the training dataset consists of both supervised and unsupervised data. VORTEX 1) extends this consistency training paradigm to a broader family of acquisition-based perturbations, 2) exhaustively studies how this framework can be leveraged for *both* image and acquisition-based augmentations, and 3) proposes a curriculum learning strategy to gradually increase reconstruction difficulty.

**Robustness to Motion.** Noise2Recon proposes a novel consistency framework for semi-supervised MRI reconstruction but solely focuses on applications to noise artifacts. While denoising is a well-known regularizer for inverse problems (Romano et al., 2017; Batson and Royer, 2019), many other acquisition-related artifacts in MRI are commonplace. In VORTEX, we explore the utility of motion augmentations as 1) a regularizer to improve robustness in label-limited settings and 2) a method to increase robustness to OOD motion artifacts. We demonstrate that motion artifact removal is as effective of a regularizer as denoising (Tables 1 and 2).

**Composing Augmentations for Multi-Artifact Correction.** Existing MRI artifact correction or removal methods, including Noise2Recon, separately handle reconstruction and artifact removal tasks, are limited to correcting for a single artifact, or require multiple unique workflows to correct for different artifacts (Usman et al., 2020; Lu et al., 2009; Jin et al., 2017). However, in practice, effects of multiple acquisition-related artifacts can be compounded even in accelerated MRI. Thus a unified framework for removing these artifacts is desirable. VORTEX establishes a framework for both image-based and acquisition-based augmentations that can be utilized to jointly reconstruct and remove multiple artifacts with a single approach.

**Curriculum Learning for Augmentations.** VORTEX extends basic consistency training to include a scheduling protocol for increasing the difficulty of augmentations over the training cycle. Results demonstrate that designing curricula for augmentations in the consistency framework can lead to considerable performance improvements in OOD settings without losing performance among in-distribution scans (Table 5). Such curricula can be helpful for joint optimization of both artifactual and artifact-free images, particularly when example difficulty is extensive (Bengio et al., 2009).

### B.3. Summary of Technical Contributions

In this work, we characterize the interface between physics-based MRI acquisition-motivated and image-based augmentations to 1) reduce label dependency and 2) increase robustness to clinically-relevant distribution shifts that are pervasive during MRI acquisition. We extend the semi-supervised consistency framework in Noise2Recon to handle both acquisition and image based perturbations in a way that is motivated by the physics-driven forward model of MRI acquisition. To ensure that we are inclusive of a broader family of acquisition-based perturbations than was available in Noise2Recon, we propose extending the semi-supervised consistency framework to handle motion, a common artifact in MRI. We exhaustively study the interaction between physics/acquisition based and image based augmentations in both fully supervised training with augmentations and semi-supervised training with the proposed consistency.

## Appendix C. Equivariant and Invariant Transforms

In this section, we provide an extended discussion of the choice of and interaction between equivariant and invariant transforms.

**Equivariance.** We simplify the precise definition of equivariance that requires group theory (Celledoni et al., 2021) to denote $f_\theta(g(x)) = g(f_\theta(x))$ for all $g \in T$ where $T$ is the set of data augmentation transformations. Intuitively, if a trained model $f_\theta$ is equivariant to a transformation $g$, then the transformation of the input directly corresponds to the transformation of the model output.

**Invariance.** Similarly, we simplify the definition of invariance to $f_\theta(g(x)) = f_\theta(x)$ for all $g \in T$ where $T$ is the the set of transformations we use for data augmentation. Intuitively, $f_\theta$ is invariant to transformation $g$ if the output of the model does not change upon applying $g$ to the input. Details on how these definitions motivate the structure of augmentations in VORTEX are provided below.

**Choosing Equivariance or Invariance.** It is important to note that, practically, specifying to which transforms the model should be equivariant or invariant is a design choice and often task-dependent. In the case of MRI, image-based augmentations proposed in MRAugment are meant to simulate differences in patient positioning and spatial scan parameters (e.g. field-of-view, nominal resolution). The differences are typically prescribed at scan time (i.e. scan parameters) or are correctable prior to the scan. In contrast, motion and noise are perturbations that occur *during acquisition*, and therefore cannot be corrected a priori. Thus, building networks that are invariant to these perturbations are critical. Based on this paradigm of transforms in MRI, spatial image transforms are classified as equivariant transforms while the physics-based transforms we propose are classified as invariant transforms.

**Composing Transforms (Extended).** Section 4.1 introduces the intuition for equivariant and invariant transformations. In this section, we formalize how transforms from these families are composed.

Let $g_1, \ldots, g_K$ be an ordered sequence of unique transforms sampled from a set of transforms $T$. Let $\bar{G}_E, \bar{G}_I$ be the sequence of sampled equivariant and invariant transforms, respectively. Thus, $\bar{G}_E = (g_i \text{ if } g_i \in T_E \; \forall \, i = 1, \ldots, K)$ (similarly for $\bar{G}_I$). Let $G_E$ and $G_I$ be the compositions of each transform in $\bar{G}_E, \bar{G}_I$, respectively. Thus, $G_E = \bar{G}_{E_{|\bar{G}_E|}} \circ \cdots \circ \bar{G}_{E_1}$ (similarly for $G_I$).

As a design choice, we select all physics-driven, acquisition-related transforms to be in the family of invariant transforms. This choice is made to ensure reconstructions are invariant to plausible acquisition-related perturbations. Thus, the family of physics-driven transforms are synonymous with the family of invariant transforms for our purposes.

Because signal from physics-driven perturbations (noise and motion) is sampled at acquisition, these perturbations are applied after undersampling in the supervised augmentation methods, where fully-sampled data is available.

## Appendix D. Experimental Details

### D.1. Dataset

In this section, we provide details for the two datasets used in this study: the mridata 3D FSE knee dataset and the fastMRI 2D multi-coil brain dataset.

#### D.1.1. MRIDATA 3D FSE KNEE DATASET

**Dataset Splits.** The mridata 3D FSE knee dataset consists of 6080 fully-Cartesian-sampled knee slices (19 scans) from healthy participants. The dataset was randomly partitioned into 4480 slices (14 scans) for training, 640 slices (2 scans) for validation, and 960 slices (3 scans) for testing.

**Simulating Data-Limited and Label-Limited Settings.** In this study, we evaluate all methods in the data-limited and label-limited regimes, where supervised examples are scarce compared to unsupervised (undersampled) examples. To simulate this scenario, a subset of training scans are retrospectively undersampled using fixed undersampling masks, resulting in unsupervised training

examples. To limit the total (supervised and unsupervised) amount of available training data, we train with only 6 of the 14 training scans, where 1 scan is supervised and 5 scans are unsupervised.

**K-space Hybridization and Sensitivity Maps.** 3D FSE scans were acquired in 3D, resulting in Fourier encoded signal along all dimensions ($k_x \times k_y \times k_z$). Because the readout dimension $k_x$ is fully-sampled in these scans, scans were decoded along the $k_x$ dimension, resulting in a hybridized k-space as mentioned in Section 4. All sensitivity maps were estimated with JSENSE as implemented in SigPy (Ong and Lustig, 2019), with a kernel width of 8 and a 20×20 center k-space auto-calibration region.

**Mask Generation.** Scans for training and evaluation were undersampled using 2D Poisson Disc undersampling, a compressed sensing-motivated pattern for 3D Cartesian imaging. Given an acceleration rate $R$, undersampling masks were generated in the $k_y \times k_z$ dimensions for all scans such that the number of pixels sampled would be approximately $\frac{|k_y||k_z|}{R}$. To maintain consistency with generated sensitivity maps, a 20×20 center k-space auto-calibration region was used when constructing undersampling masks for all examples. To simulate prospectively undersampled acquisitions, scans were retrospectively undersampled with a fixed 2D Poisson Disc undersampling pattern (Bridson, 2007). Following Cartesian undersampling convention, all $k_y \times k_z$ slices for a single scan are undersampled with the identical 2D Poisson Disc mask. This procedure was used for both simulating prospectively undersampled scans during training (i.e. unsupervised examples) and evaluation. All undersampling masks for testing and simulating undersampled k-space are generated with an unique, fixed random seed for each scan to ensure reproducibility.

### D.1.2. FASTMRI BRAIN MULTI-COIL DATASET

**Dataset Splits.** The distributed validation split of the fastMRI 2D brain multi-coil dataset was divided into 757 scans for training, 207 scans for validation, and 414 scans for testing. To control for confounding variables when comparing performance between reconstruction methods, all data splits were filtered to include only T2-weighted scans acquired at a 3T field strength, resulting in 266, 70, and 137 scans for training, validation, and testing, respectively. Data-limited and label-limited training settings were simulated by limiting training data to 18 supervised and 36 unsupervised scans and validation data to 50 scans.

**Sensitivity Maps.** Like for mridata, sensitivity maps were estimated using JSENSE with a kernel width of 8 and calibration region of 12×12. This calibration region corresponds to the 4% auto-calibration region used for 8x undersampling.

**Mask Generation.** Scans for training and evaluation were undersampled using 1D random undersampling, a compressed sensing-motivated pattern for 2D Cartesian imaging (Lustig et al., 2007). Given an acceleration rate $R$, undersampling masks were generated in the $k_y$ phase-encode dimension for all scans such that the number of readout lines sampled would be approximately $\frac{|k_y|}{R}$. Training and evaluation was conducted at $R = 8$ acceleration with a 4% auto-calibration region. Like in mridata, fixed undersampling masks were generated to simulate prospectively undersampled data and for the testing data to ensure reproducibility.

### D.2. Baselines

We compared VORTEX to state-of-the-art supervised, supervised augmentation, and self-supervised MRI reconstruction baselines. We provide an overview of these methods and their notation in the main text. Hyperparameters for all methods are provided in Appendix D.3.1.

**Supervised.** We compared VORTEX to standard supervised training without augmentations (termed *Supervised*). In supervised training, fully-sampled scans are retrospectively undersampled. The model is trained to reconstruct the fully-sampled scan from its undersampled counterpart. Note, in supervised settings, only fully-sampled scans can be used for training. Any prospectively undersampled (unsupervised) scans cannot be leveraged in this setup.

**Supervised+Augmentation (*Aug*) and MRAugment.** Supervised baselines with augmentation (termed *Aug*) were trained with image and/or physics-based augmentations, which are denoted by parentheses. Image-based augmentations were applied prior to the retrospective undersampling, following the MRAugment protocol. Physics-based acquisition augmentations were applied after this undersampling to model the MRI data acquisition process. For example *Aug (Motion)* indicates a supervised method trained with motion augmentations. Image-based augmentations were identical to those used in MRAugment. As such, *Aug (Image)* is equivalent to MRAugment, and is referred to as such for readability.

**SSDU.** We also compared VORTEX to the state-of-the-art self-supervised learning via data undersampling (SSDU) baseline (Yaman et al., 2020). This method was originally proposed for fully unsupervised learning, in which all training scans are prospectively undersampled. We propose a trivial extension to adapt it for the semi-supervised setting. In cases of prospectively undersampled (unsupervised) data, the training protocol proposed in SSDU was used. Fully-sampled (supervised) data was retrospectively undersampled using the undersampling method and acceleration for the specified experiment. These simulated undersampled scans were used as inputs to the SSDU protocol. Because the retrospective undersampling is done dynamically (i.e. each time a supervised example is sampled), it may serve as a method of augmenting supervised scans.

**Compressed Sensing (CS).** As a non-DL baseline, we used $\ell_1$-wavelet regularized CS (Lustig et al., 2007). Note, because of the time-intensive nature of CS-based reconstructions, CS is difficult to scale for reconstructing large datasets like the fastMRI brain dataset. Thus, we only evaluate CS on the mridata 3D FSE knee dataset.

### D.3. Training Details

All code is written in Python with PyTorch 1.6 and is available at `https://github.com/ad12/meddlr`.

#### D.3.1. HYPERPARAMETERS

**Architecture and Optimization.** All models used a 2D U-Net architecture with (Ronneberger et al., 2015) with 4 pooling layers. Convolutional block at depth $d$ consisted of two convolutional layers with $32^d$ channels for $d = \{1, \ldots, 5\}$. All models were trained with the Adam optimizer with default parameters ($\beta_1$=0.9, $\beta_2$=0.999) and weight decay of 1e-4 for 200 epochs (Kingma and Ba, 2014; Loshchilov and Hutter, 2017). Training was conducted with an effective training batch size

Table 4: Data augmentation configuration for mridata 3D FSE knee dataset experiments. $p$ is the effective probability of applying an augmentation. In MRAugment, this is equivalent to the base probability multiplied by the weighting factor. Acquisition-based augmentations were configured in separate experiments at both light and heavy settings.

| Kind | Transform | Parameters | p |
|---|---|---|---|
| | H-Flip | N/A | 0.275 |
| | V-Flip | N/A | 0.275 |
| | $k \times 90°$ rotation | $k \in \{2\}$ | 0.275 |
| Image | Rotation | [-180°, 180°] | 0.275 |
| | Translation | [-10%, 10%] | 0.55 |
| | Scale | [0.75, 1.25] | 0.55 |
| | Shear | [-15°, 15°] | 0.55 |
| Acquisition | Gaussian Noise | $\sigma$=[0.1,0.3] (light) $\sigma$=[0.2,0.5] (heavy) | 0.2 |
| | Motion | $\alpha$=[0.1,0.3] (light) $\alpha$=[0.2,0.5] (heavy) | 0.2 |

of 24 and learning rate $\eta$=1e-3. All models used VORTEX methods used 1:1 balanced sampling between supervised and unsupervised examples (Desai et al., 2021a).

***Aug* Baselines and MRAugment.** Supervised augmentation baselines were trained with image-based and acquisition-based augmentations. Image-based augmentations for each dataset followed the augmentation configuration provided in the MRAugment. With the mridata 3D FSE knee dataset, integer rotations could only be conducted at 180 degrees due to the anisotropic matrix shape of the $ky \times kz$ slice. *Aug* baselines using physics-driven acquisition-based augmentations used a maximum probability of $p = 0.2$ as recommended by Desai et al. (2021a), and use the same range of $\sigma$ for noise and $\alpha$ for motion that are used in the corresponding VORTEX experiments. Augmentations, their parameters, and their effective probabilities used for the mridata 3D FSE knee dataset are listed in Table 4. All augmentation methods were trained with the exponential augmentation probability scheduler with $\gamma = 5$ and a scheduling period equivalent to the training length as proposed by Fabian et al. (2021).

**SSDU.** SSDU is sensitive to the loss function and masking extent ($\rho$). Thus, these hyperparameters that should be optimized for different datasets. We swept through loss functions k-space $\ell_1$, k-space $\ell_1$-$\ell_2$, and image $\ell_1$ and masking extent $\rho = 0.2, 0.4, 0.6$. Models with the highest validation cPSNR were selected for all SSDU experiments. For the mridata 3D FSE knee dataset, the configuration with loss function k-space $\ell_1$ and $\rho = 0.4$ was used. For the fastMRI multi-coil brain dataset, the configuration with normalized $\ell_1$-$\ell_2$ loss in k-space and $\rho = 0.2$ was used.

**Compressed Sensing (CS).** Because CS is sensitive to the regularization parameter $\lambda$, this parameter must be carefully tuned for different motion and noise settings. To hand-tune $\lambda$ for both light and heavy motion levels, we swept through $\lambda$ values within the range $[0, 0.5]$ and selected the optimal $\lambda$ that balances reconstruction quality and artifact mitigation. For the 3D FSE mridata knee dataset, $\lambda = 0.1$ was used for light motion, and $\lambda = 0.15$ was used for heavy motion at 12x and 16x

Table 5: Comparison of different scheduling methods and warmup periods on the mridata knee multi-coil dataset with heavy motion augmentations. All scheduling methods outperform non-scheduled training (base). There is no advantage of a specific scheduling protocol, suggesting that some curriculum is better than none.

| Perturbation | None | | Motion (light) | | Motion (heavy) | |
|---|---|---|---|---|---|---|
| Curricula | SSIM | cPSNR (dB) | SSIM | cPSNR (dB) | SSIM | cPSNR (dB) |
| None | 0.861 | 36.4 | 0.855 | 35.8 | 0.819 | 33.2 |
| Linear (20e) | 0.866 | 36.4 | 0.862 | 35.8 | **0.828** | 33.3 |
| Linear (100e) | 0.877 | 36.3 | **0.871** | 35.8 | 0.822 | 32.6 |
| Linear (200e) | 0.869 | 36.4 | 0.865 | 35.8 | 0.817 | 32.7 |
| Exp (20e, $\gamma = 5$) | 0.865 | **36.4** | 0.857 | **35.9** | 0.822 | **33.4** |
| Exp (100e, $\gamma = 5$) | 0.864 | 36.3 | 0.857 | 35.8 | 0.812 | 33.2 |
| Exp (200e, $\gamma = 5$) | **0.877** | 36.4 | 0.867 | 35.8 | 0.812 | 32.3 |

acceleration. For in-distribution (no noise, no motion), light noise, and heavy noise settings, we use the regularization parameters suggested by Desai et al. (2021a).

**Consistency Augmentations in VORTEX.** Like *Aug* baselines, VORTEX was trained with combinations of image and physics-based augmentations. We use the same parenthetical nomenclature to indicate the augmentation type used in the consistency branch (e.g. *VORTEX (Motion)* for motion consistency). The family of image augmentations used for consistency in VORTEX were identical to those used in MRAugment. Physics-based consistency augmentations were sampled from either the light ($\mathcal{R}(\cdot)$=[0.1, 0.3)) or heavy ($\mathcal{R}(\cdot)$=[0.2, 0.5)) range during training.

### D.4. Evaluation

D.4.1. EVALUATION SETTINGS

We perform evaluation in both in-distribution and clinically-relevant, simulated OOD settings. In-distribution evaluation consisted of evaluation on the test set described in D.1.

For OOD evalution, we considered two critical settings that have been shown to affect image quality: (1) decrease in SNR and (2) motion corruption. The extent of the distribution shift is synonymous with the difficulty level for each perturbation ($\sigma$ for noise, $\alpha$ for motion), where larger difficulty levels indicate correspond to larger shifts. Thus, we define *low* and *heavy* noise and motion difficulty levels for evaluation – low noise $\sigma$=0.2, heavy noise $\sigma$=0.4, low motion $\alpha$=0.2, heavy motion $\alpha$=0.4. These values are selected based on visual inspection of clinical scans (see 4.1.1). Note, by definition ($\sigma$=0, $\alpha$=0) corresponds to the in-distribution evaluation.

Given difficulty levels for motion and noise, each scan was perturbed by a noise or phase error (motion) maps generated with a set difficulty level. These perturbations were fixed for each testing scan to ensure reproducibiilty and identical perturbations in the test set across different experiments.

In the text, we refer to different evaluation configurations as *perturbations*. *None* indicates the in-distribution setting. *LN*, *HN*, *LM*, *HM* correspond to light noise, heavy noise, light motion, and heavy motion, respectively.

Table 6: Impact of training duration on cPSNR of supervised methods without augmentations (*Supervised*), supervised methods with motion augmentations (*Aug (Motion)*), MRAugment, and VORTEX with motion consistency (*VORTEX (Motion)*). Training duration are percentages of the full training duration (200 epochs). Asterisk (*) indicates the default training configuration. Both supervised augmentation methods and MRAugment are more sensitive to training time than Supervised or VORTEX methods. Supervised underperforms Aug, MRAugment, and VORTEX. VORTEX achieves highest performance and is insensitive to training duration relative to the other methods.

| | | Perturbation | |
|---|---|---|---|
| Model | None | Motion (light) | Motion (heavy) |
| Supervised (10%) | 35.0 | 33.3 | 27.4 |
| Supervised (25%) | 35.3 | 32.3 | 27.1 |
| Supervised (50%) | 35.5 | 32.0 | 26.4 |
| Supervised (100%)* | 35.8 | 33.6 | 27.0 |
| Supervised (200%) | 36.0 | 33.9 | 27.6 |
| Supervised (300%) | 36.0 | 33.9 | 27.6 |
| MRAugment (10%) | 35.4 | 32.3 | 26.0 |
| MRAugment (25%) | 35.8 | 31.5 | 25.1 |
| MRAugment (50%) | 36.0 | 31.5 | 24.3 |
| MRAugment (100%) | 36.2 | 31.8 | 24.0 |
| MRAugment (200%) | 36.3 | 32.2 | 24.3 |
| MRAugment (300%) | 36.4 | 33.4 | 25.0 |
| Aug (Motion) (10%) | 34.8 | 33.9 | 30.8 |
| Aug (Motion) (25%) | 35.3 | 34.5 | 31.4 |
| Aug (Motion) (50%) | 35.4 | 34.6 | 31.1 |
| Aug (Motion) (100%)* | 35.9 | 35.1 | 31.5 |
| Aug (Motion) (200%) | 36.0 | 35.1 | 30.8 |
| Aug (Motion) (300%) | 36.0 | 35.2 | 32.1 |
| VORTEX (Motion) (10%) | 36.2 | 35.5 | 32.4 |
| VORTEX (Motion) (25%) | 36.3 | 35.7 | 33.1 |
| VORTEX (Motion) (50%) | 36.4 | 35.8 | 33.2 |
| VORTEX (Motion) (100%)* | 36.4 | 35.8 | 33.2 |
| VORTEX (Motion) (200%) | 36.3 | 35.7 | 33.0 |
| VORTEX (Motion) (300%) | 36.3 | 35.7 | 33.0 |

### D.4.2. METRIC SELECTION

Conventional computational imaging uses magnitude metrics for quantifying image quality. However, MRI images contain both magnitude and phase information (i.e. real and imaginary components). Because phase-related errors may not be captured by magnitude metrics, we use a combination of complex and magnitude metrics – complex PSNR (cPSNR) and magnitude SSIM – to quantify image quality. Equation 1 defines the cPSNR formulation for complex-valued ground truth $x_{ref}$ and prediction $x_{pred}$. $|| \cdot ||_2$ corresponds to the complex-$\ell_2$ norm and $| \cdot |$ denotes the magnitude of complex-valued input. Additionally, SSIM has shown to be a better corollary for MRI reconstruction quality compared to pSNR on magnitude images (Knoll et al., 2020). Thus, we use SSIM to quantify magnitude image quality.

$$\text{cPSNR (dB)} = 20 \log_{10} \frac{\max |x_{ref}|}{||x_{pred} - x_{ref}||_2} \tag{1}$$

Table 7: Ablation for acceleration – 12x vs 16x. Similar to results at 16x acceleration, *VORTEX (Motion)* outperformed supervised methods, and MRAugment at 12x acceleration. This may suggest that VORTEX is broadly applicable to different acceleration levels.

| | Perturbation | None | | Motion (light) | | Motion (heavy) | |
|---|---|---|---|---|---|---|---|
| Aug Range | Model | SSIM | cPSNR (dB) | SSIM | cPSNR (dB) | SSIM | cPSNR (dB) |
| N/A | Supervised 12x | 0.814 | 36.2 | 0.814 | 32.4 | 0.689 | 25.4 |
| | Supervised 16x | 0.798 | 35.8 | 0.809 | 33.6 | 0.706 | 27.0 |
| N/A | MRAugment 12x | 0.828 | 36.5 | 0.814 | 31.9 | 0.637 | 23.6 |
| | MRAugment 16x | 0.811 | 36.2 | 0.793 | 31.8 | 0.660 | 24.0 |
| N/A | SSDU 12x | 0.819 | 34.9 | 0.816 | 34.5 | 0.762 | 30.9 |
| | SSDU 16x | 0.787 | 34.9 | 0.783 | 34.7 | 0.734 | 31.9 |
| light | Aug (Motion) 12x | 0.811 | 36.1 | 0.807 | 35.3 | 0.765 | 31.3 |
| | Aug (Motion) 16x | 0.802 | 35.6 | 0.793 | 34.7 | 0.739 | 30.4 |
| heavy | Aug (Motion) 12x | 0.818 | 36.1 | 0.811 | 35.2 | 0.758 | 31.2 |
| | Aug (Motion) 16x | 0.793 | 35.9 | 0.793 | 35.1 | 0.751 | 31.5 |
| light | VORTEX Motion 12x | 0.881 | 36.8 | 0.875 | 36.1 | 0.815 | 32.1 |
| | VORTEX Motion 16x | 0.882 | 36.4 | 0.875 | 35.7 | 0.813 | 31.5 |
| heavy | VORTEX Motion 12x | 0.888 | 36.7 | 0.883 | 36.1 | 0.846 | 33.5 |
| | VORTEX Motion 16x | 0.861 | 36.4 | 0.855 | 35.8 | 0.819 | 33.2 |

Table 8: Pixel-level vs. latent space consistency. **LM**: light motion; **HM**: heavy motion.

| Model | cPSNR (dB) | SSIM | cPSNR (dB) (LM) | SSIM (LM) | cPSNR (dB) (HM) | SSIM (HM) |
|---|---|---|---|---|---|---|
| Supervised | 35.8 | 0.798 | 33.6 | 0.809 | 27.1 | 0.706 |
| Pixel-Level | 36.4 | 0.873 | 35.9 | 0.866 | 33.2 | 0.828 |
| $R_4$ | 36.4 | 0.877 | 34.7 | 0.865 | 29.8 | 0.778 |
| $R_3,R_4$ | 36.4 | 0.873 | 34.0 | 0.852 | 30.1 | 0.781 |
| $R_2,R_3,R_4$ | 36.3 | 0.873 | 34.4 | 0.854 | 29.5 | 0.769 |
| $R_1,R_2,R_3,R_4$ | 36.3 | 0.875 | 34.7 | 0.864 | 30.3 | 0.775 |

By default, metrics were computed over the full 3D scan. An additional set of metrics were also computed per reconstructed slice (termed *slice metrics*). Because different slices have different extents of relevant anatomy, per-slice metrics can provide a more nuanced comparison of 2D slice reconstructions among different methods. Statistical comparisons were conducted using Kruskal-Wallis tests and corresponding Dunn posthoc tests with Bonferroni correction ($\alpha$=0.05). All statistical analyses were performed using the SciPy library.

## Appendix E. Ablations

We perform ablations to understand two design questions for key components in our framework: (1) Can consistency be enforced at different points in the network; (2) How should example difficulty be specified during training. All methods use the default configurations specified in Appendix D.3. To evaluate each piece thoroughly, we consider augmentation and VORTEX approaches trained with heavy motion perturbations.

### E.1. Latent Space vs Pixel-level Consistency

We compare enforcing consistency at the pixel-level output image versus learned latent representations at varying U-Net resolution levels. Let $R_k$ be $k^{th}$ resolution level at which consistency is enforced, where $k \in \{1, 2, 3, 4\}$ since our U-Net architecture had 4 pooling layers. For the $k^{th}$ resolution level, we enforce consistency after the final convolution in the encoder, and after the transpose convolution in the decoder. For $k = 4$, consistency is enforced at the bottleneck layer, after the convolution in the encoder. To control for the impact of loss weighting, we normalize $\lambda$ by the number of consistency losses that

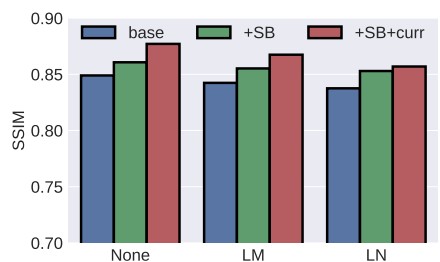

Figure 4: Ablation for balanced sampling (*SB*) and augmentation curricula (*curr*) in *VORTEX (Motion)*.

are computed in latent space when consistency is enforced at multiple resolution levels $R_k$. We compare these approaches in the case of light and heavy motion.

We find that latent space consistency performed similarly across all resolution levels, outperforming the *Supervised* baseline on both in- and out-of-distribution data (Table 8). For in-distribution data, latent space consistency at any resolution level performed on par with pixel-level consistency. However, for OOD data, it performed considerably worse than pixel-level consistency, by at least 0.047 SSIM and 3.2dB cPSNR under heavy motion. Although not common in the consistency training literature, we find that pixel-level consistency was a better technique for capturing the semantics of global distribution shifts such as motion for accelerated MRI reconstruction, which might occur at the pixel-level.

### E.2. Augmentation Scheduling.

We seek to quantify the utility of scheduling augmentation difficulty in VORTEX's consistency branch (see §4.2). We evaluate linear and exponential scheduling functions with different warm up schedules – 10%, 50%, and 100% of the training period. We show that curricula methods outperformed non-curricula methods for both in-distribution and OOD evaluation (Table 5 in the Appendix). However, no one curricula configuration outperformed others, which may indicate that all curricula methods are feasible ways to schedule augmentations. Curriculum learning is also compatible with the balanced sampling protocol proposed by Desai et al. (2021a), where supervised and unsupervised examples are sampled at a fixed ratio during training.

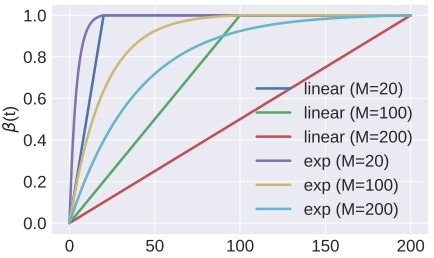

Figure 5: Augmentation difficulty curricula over training period of 200 epochs using linear and exponential (*exp*) schedulers defined in §4.2. Time constant for exponential scheduling $\tau = \frac{M}{\gamma}$ where $\gamma$=5.

Incorporating balanced sampling (SB) into training led to an increase in SSIM for both in-distribution and OOD light motion and light noise evaluation configurations (Fig. 4). Increase in SSIM may indicate that curricula can help the network gradually learn useful representations without a mode collapse into the trivial solution (i.e. image blurring), which is common for pixel-level losses.

Table 9: Average performance on mridata 3D FSE knee dataset with light motion ($\alpha$=0.2) and noise ($\sigma$=0.2) perturbations. Physics augmentations are compositions of noise and motion in their *heavy* ($\mathcal{R}(\alpha) = \mathcal{R}(\sigma) = [0.2, 0.5]$) training difficulty configurations. Models are identical to those reported in Table 1.

| Perturbation | Motion (light) | | Noise (light) | |
|---|---|---|---|---|
| | SSIM | cPSNR (dB) | SSIM | cPSNR (dB) |
| Compressed Sensing | 0.810 (0.024) | 29.8 (1.7) | 0.828 (0.002) | 32.9 (0.2) |
| Supervised | 0.809 (0.028) | 33.6 (0.2) | 0.830 (0.024) | 33.8 (0.3) |
| MRAugment | 0.793 (0.021) | 31.8 (1.8) | 0.793 (0.024) | 33.0 (0.4) |
| SSDU | 0.783 (0.025) | 34.7 (0.2) | 0.752 (0.024) | 33.7 (0.3) |
| Aug (Physics) | 0.785 (0.036) | 35.0 (0.4) | 0.761 (0.049) | 34.2 (0.3) |
| Aug (Image+Physics) | 0.782 (0.045) | **35.5 (0.4)** | 0.750 (0.051) | 34.6 (0.5) |
| VORTEX (Image) | 0.831 (0.036) | 32.5 (0.8) | **0.859 (0.004)** | 33.6 (0.1) |
| VORTEX (Physics) | 0.846 (0.025) | 35.2 (0.4) | 0.841 (0.035) | **34.7 (0.2)** |
| VORTEX (Image+Physics) | **0.849 (0.024)** | 35.0 (0.5) | 0.850 (0.028) | 34.3 (0.1) |

### E.3. Training Time

We ablate the sensitivity of the performance of supervised, augmentation, and VORTEX methods to training duration. To compute the performance at different training duration, we select best checkpoints (quantified by validation cPSNR) up to a given duration and run evaluation using these weights. As all methods were trained for 200 epochs, we compare the performance at training times of 10% (20 epochs), 25% (50 epochs), 50% (100 epochs), and 100% (200 epochs). Supervised methods were insensitive to training time, but considerably underperformed both supervised augmentation (Aug) and VORTEX (Table 6). Augmentation based methods were sensitive to training time, with changes in cPSNR of >1dB. VORTEX achieved the highest performance across all metrics and evaluation setups and was relatively insensitive to training duration.

### E.4. Sensitivity to Acceleration Factors

We evaluated the performance of VORTEX at different acceleration factors in Table 7. At 12x acceleration, VORTEX trained with heavy motion recovered +0.061 SSIM and +0.8dB cPSNR compared to the *Supervised* baseline in the in-distribution setting. At the same acceleration, VORTEX also outperformed the *Supervised* baseline by +0.157 SSIM and +8.1dB cPSNR. The stability of VORTEX at different accelerations may indicate that VORTEX is generalizable across different acceleration extents.

## Appendix F. Extended Results

In this section, we provide extended results for the mridata dataset using slice metrics and for the fastMRI multi-coil brain dataset (Zbontar et al., 2018).

### F.1. Extended mridata Results

**Light Perturbations.** Table 9 shows performance of baselines and VORTEX on the mridata knee dataset with light motion ($\alpha$=0.2) and noise ($\sigma$=0.2) perturbations. Like in heavy settings (Table 1), VORTEX methods consistently achieve higher SSIM than and comprable cPSNR to baselines. Improved SSIM values may indicate VORTEX can recover structural features in the image in cases

Table 10: Slice metrics (mean [standard deviation]) on the mridata knee dataset. Asterisk (*) indicates significant performance of VORTEX over *all* baselines ($p < 0.05$).

| Perturbation | None | | Motion (heavy) | | Noise (heavy) | |
|---|---|---|---|---|---|---|
| Model | SSIM | cPSNR (dB) | SSIM | cPSNR (dB) | SSIM | cPSNR (dB) |
| Compressed Sensing | 0.682 (0.122) | 29.4 (3.2) | 0.559 (0.135) | 18.9 (2.4) | 0.534 (0.102) | 24.8 (2.5) |
| Supervised | 0.635 (0.133) | 29.7 (3.7) | 0.545 (0.117) | 21.4 (2.5) | 0.591 (0.139) | 25.8 (2.9) |
| MRAugment | 0.653 (0.130) | 30.1 (3.5) | 0.505 (0.106) | 18.6 (2.3) | 0.563 (0.128) | 25.0 (2.8) |
| SSDU | 0.621 (0.147) | 28.9 (3.4) | 0.564 (0.146) | 25.9 (3.5) | 0.528 (0.142) | 26.7 (2.8) |
| Aug (Physics) | 0.623 (0.144) | 29.6 (3.6) | 0.566 (0.136) | 26.0 (3.8) | 0.557 (0.144) | 27.6 (3.1) |
| Aug (Image+Physics) | 0.618 (0.136) | 30.1 (3.4) | 0.565 (0.134) | **26.9 (3.8)** | 0.540 (0.134) | 27.9 (2.8) |
| VORTEX (Image) | 0.718 (0.125)* | **30.4 (3.4)** | 0.499 (0.110) | 20.6 (2.3) | 0.584 (0.104) | 25.8 (2.5) |
| VORTEX (Physics) | **0.729 (0.138)*** | 30.3 (3.4) | **0.628 (0.137)*** | 26.0 (3.8) | 0.653 (0.143)* | **28.1 (2.8)** |
| VORTEX (Image+Physics) | 0.716 (0.131)* | 30.3 (3.4) | 0.616 (0.130)* | 25.3 (3.7) | **0.658 (0.132)*** | 27.5 (2.7) |

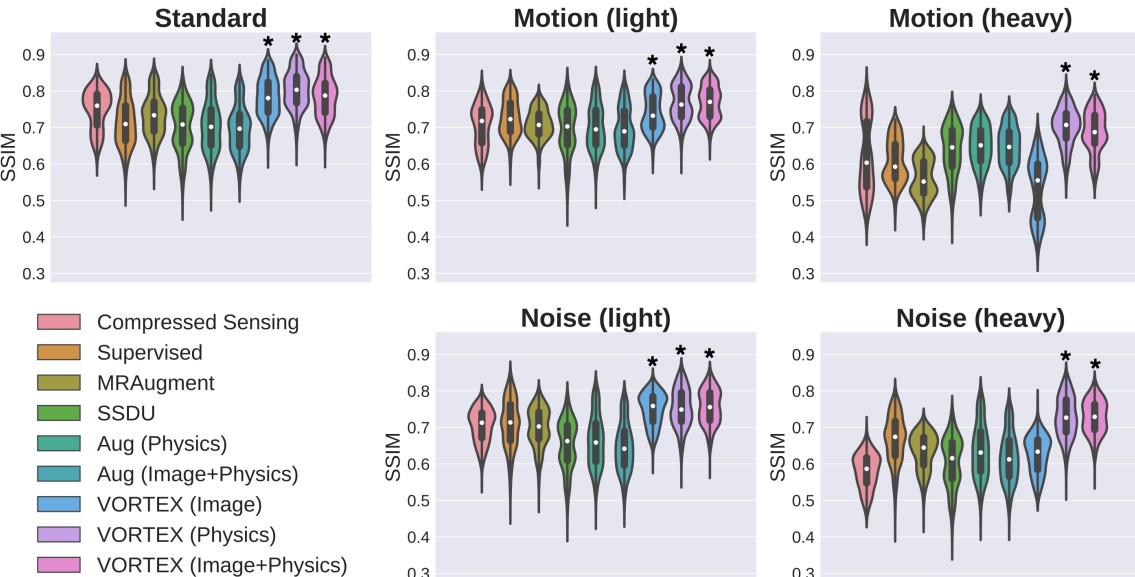

Figure 6: SSIM at different perturbation levels on center 50% of slices. Asterisk (*) indicates significant performance (p<0.05) over *all* baselines (Compressed Sensing, *Supervised*, MRAugment, SSDU, *Aug*). VORTEX methods significantly outperformed baselines in both in-distribution and OOD perturbation settings. VORTEX also had less variance across slices, which may indicate more consistent per-slice reconstruction than baseline methods.

of varying extents of perturbations. This trait may be useful for generalizing to data with unknown perturbation extents and may reduce the need for extensive hyperparameter search.

**Slice metrics.** Table 10 shows slice metrics of baselines and VORTEX on the mridata knee dataset. Among slice metrics, VORTEX also outperforms all baselines in both in-distribution and OOD settings. In particular, VORTEX significantly outperformed all baselines in SSIM in all evaluation settings (p<0.05). This may indicate that VORTEX has higher fidelity in recovering image structure even in OOD settings where perturbations can result in a considerable degradation in SSIM. To understand the efficacy of VORTEX in anatomically-dense regions of the image, slice metrics were also computed on the center 50% of axial slices (Fig. 6). Not only does VORTEX significantly outperform all baselines (p<0.05), it also had the least variance among all reconstruction methods.

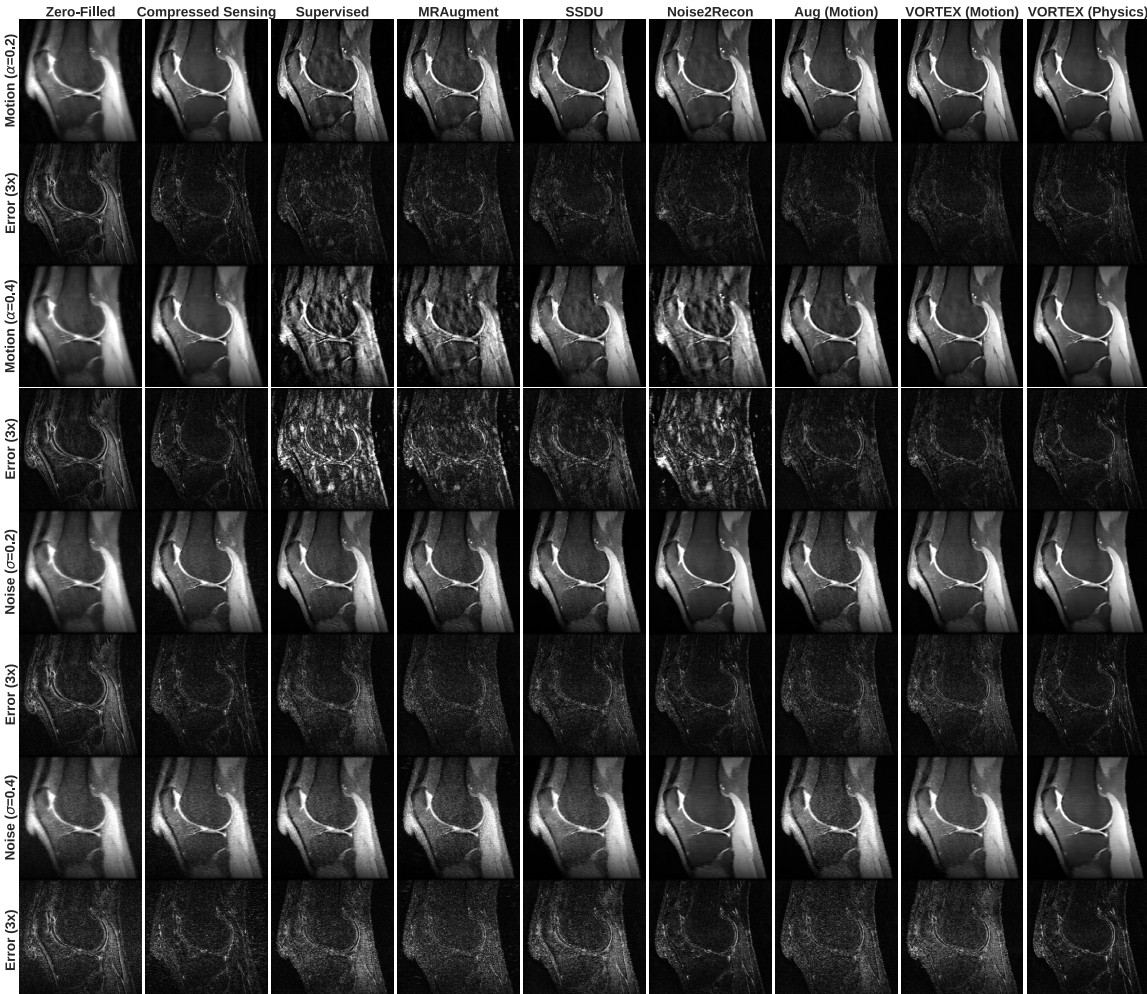

Figure 7: Sample scan reconstruction (mridata) with different extents of motion ($\alpha$) and noise ($\sigma$) perturbations. Scans were reconstructed along axial slices, but are reformatted along the sagittal direction to illustrate through-plane artifacts. Unresolved motion artifacts can result in coherent ghosting and streaking artifacts along the through-plane direction. Noise artifacts are also more prominent in fat-suppressed regions (e.g. bone) and near articular (femoral, tibial, patellar) cartilage. VORTEX suppresses both noise and motion artifacts, producing higher quality images even along reformatted directions.

This may indicate VORTEX can *consistently* reconstruct higher quality images compared to state-of-the-art supervised, augmentation-based, and self-supervised methods.

**Scan reformatting.** Reconstructed scans are often reformatted to enable anatomical inspection from multiple views. In Fig. 7, we show an example mridata 3D FSE scan, which has been reformatted to the sagittal plane. Artifacts in reconstructions from baseline compressed sensing and deep learning-based methods are acutely visible in the reformatted slice. Motion ghosting artifacts seen in the axial plane (Fig. 3) appeared as coherent streaks in the sagittal reformat. Noise artifacts were amplified by DL-based baselines and lead to blurring among compressed sensing reconstructions. In contrast, VORTEX-based reconstructions sufficiently suppressed these artifacts, resulting in high-quality reformatted images.

Table 11: Test performance (mean [standard deviation]) on the fastMRI multi-coil brain dataset at 8x acceleration. Results are shown on both in-distribution data and different motion levels of $\alpha = 0.4, 0.6, 0.8, 1.0$ for Supervised, SSDU, MRAugment, augmentation baselines, and VORTEX.

| Perturbation | None | | Motion ($\alpha = 0.4$) | | Motion ($\alpha = 0.6$) | | Motion ($\alpha = 0.8$) | | Motion ($\alpha = 1.0$) | |
|---|---|---|---|---|---|---|---|---|---|---|
| Model | SSIM | cPSNR (dB) | SSIM | cPSNR (dB) | SSIM | cPSNR (dB) | SSIM | cPSNR (dB) | SSIM | cPSNR (dB) |
| Supervised | 0.811 (0.052) | 27.4 (2.6) | 0.681 (0.108) | 21.6 (3.2) | 0.594 (0.161) | 18.8 (3.7) | 0.537 (0.172) | 16.7 (3.9) | 0.522 (0.160) | 15.2 (3.8) |
| MRAugment | **0.852 (0.037)** | 29.2 (1.4) | 0.731 (0.097) | 21.9 (3.6) | 0.641 (0.164) | 18.7 (4.1) | 0.578 (0.182) | 16.3 (4.3) | 0.564 (0.171) | 14.7 (4.0) |
| SSDU | 0.844 (0.042) | 27.9 (1.3) | 0.798 (0.081) | 22.8 (3.4) | 0.704 (0.178) | 19.6 (4.3) | 0.622 (0.208) | 17.0 (4.6) | 0.592 (0.197) | 15.2 (4.4) |
| Noise2Recon ($\mathcal{R}(\sigma) = [0.5, 0.7]$) | 0.842 (0.044) | **29.7 (1.5)** | 0.736 (0.102) | 22.2 (3.7) | 0.642 (0.168) | 18.9 (4.2) | 0.577 (0.181) | 16.4 (4.4) | 0.558 (0.168) | 14.7 (4.1) |
| Aug (Motion, $\mathcal{R}(\alpha) = [0.2, 0.5]$) | 0.816 (0.048) | 28.1 (1.5) | 0.782 (0.084) | 24.2 (2.8) | 0.687 (0.189) | 21.0 (4.2) | 0.607 (0.224) | 18.2 (5.0) | 0.589 (0.207) | 16.1 (5.2) |
| Aug (Motion, $\mathcal{R}(\alpha) = [0.5, 0.7]$) | 0.809 (0.048) | 27.8 (1.4) | 0.767 (0.079) | 24.2 (2.5) | 0.678 (0.175) | 22.2 (3.8) | 0.605 (0.203) | 19.7 (4.5) | 0.582 (0.193) | 17.8 (4.5) |
| VORTEX (Motion, $\mathcal{R}(\alpha) = [0.2, 0.5]$) | 0.840 (0.045) | 29.4 (1.5) | **0.823 (0.050)** | **25.9 (2.2)** | **0.749 (0.148)** | 23.2 (3.7) | 0.688 (0.181) | 20.1 (5.2) | **0.676 (0.170)** | 17.8 (5.8) |
| VORTEX (Motion, $\mathcal{R}(\alpha) = [0.5, 0.7]$) | 0.833 (0.045) | 29.2 (1.5) | 0.781 (0.059) | 25.8 (1.9) | 0.741 (0.084) | **24.3 (2.7)** | **0.702 (0.100)** | **22.8 (3.3)** | **0.683 (0.097)** | **21.7 (3.2)** |
| VORTEX (Image+Motion, ($\alpha) = [0.5, 0.7]$) | 0.840 (0.043) | 29.0 (1.5) | 0.783 (0.064) | 24.1 (2.7) | 0.711 (0.133) | 21.9 (3.3) | 0.661 (0.146) | 19.6 (4.1) | 0.652 (0.137) | 17.7 (4.4) |
| VORTEX (Image+Physics, ($\alpha) = [0.5, 0.7]$) | 0.834 (0.043) | 29.3 (1.6) | 0.740 (0.097) | 22.2 (3.7) | 0.645 (0.168) | 18.9 (4.2) | 0.581 (0.180) | 16.4 (4.4) | 0.565 (0.168) | 14.7 (4.1) |

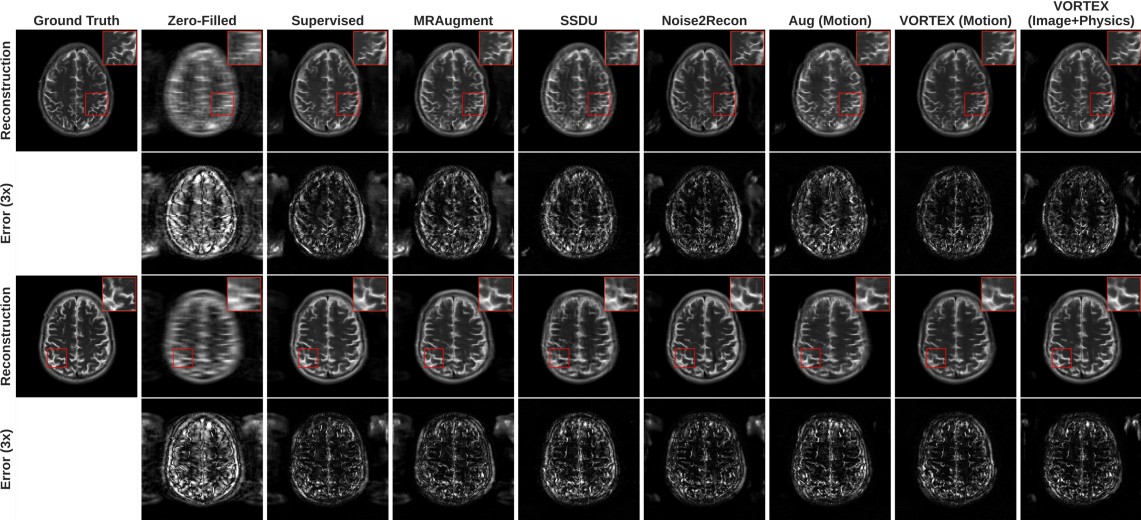

Figure 8: Sample fastMRI brain reconstructions with heavy motion ($\alpha$=0.4) perturbation. All baseline methods suffer from coherent ghosting artifacts. SSDU can suppress the coherence of these artifacts, but results in extensive blurring of vasculature. VORTEX can minimize this blurring while suppressing coherent ghosting artifacts.

## F.2. fastMRI Results

We compare VORTEX to Supervised, SSDU, Aug (Motion), and MRAugment baselines for in distribution and OOD motion settings of different motion levels on the fastMRI multi-coil brain dataset in Table 11 (Zbontar et al., 2018). Data preparation and experimental details follow the description in Appendix D, and all experiments are conducted at 8x acceleration. We demonstrate that VORTEX has comparable performance to baselines for in distribution, and outperforms SSDU by +0.025 SSIM and +3.1dB cPSNR, and MRAugment by +0.092 SSIM and 4.0dB cPSNR on motion level $\alpha = 0.4$; SSDU by +0.045 SSIM and +4.7dB cPSNR, and MRAugment by +0.108 SSIM and +5.6dB cPSNR on motion level $\alpha = 0.6$; SSDU by +0.08 SSIM and +5.8dB cPSNR, and MRAugment by +0.124 SSIM and +6.5dB cPSNR on motion level $\alpha = 0.8$; SSDU by +0.091 SSIM and +6.5dB cPSNR, and MRAugment by +0.119 SSIM and +7.0dB cPSNR on motion level $\alpha = 1.0$. This demonstrates that the effectiveness of VORTEX for both in distribution and OOD data generalizes to 2D MRI sequences which implies broader clinical utility. Sample reconstructions are shown in Fig. 8.

