# OpenReview forum: "VORTEX: Physics-Driven Data Augmentations Using Consistency Training for Robust Accelerated MRI Reconstruction"
_MIDL.io/2022/Conference — MIDL 2022_

### Official Review · Reviewer_oizx · 2022-01-24

**Confidence:** 4
**Preliminary Rating:** 4
**Recommendation:** Poster

**Summary:**

This paper aim to address the effects of motion and variable SNR-levels in MRI-acquisition, which are timely and relevant topics. To this end, the authors extend the earlier presented Noise2Recon framework that jointly reconstructs and denoises MRI-data. Futhermore data is augmented through a forward model. Detailed ablation studies are performed to assess the performance of individual components of this work.

**Strengths:**

To arrive at clinically relevant MRI-acquisition, data augmentation is performed through the forward model (section 3), targeting the multi-coil setting with coil sensitivities and motion estimation.

Affine transformations are indeed applied to augment data (Table 4)

Separate experiments on noise and motion are performed.


**Weaknesses:**

The motion model (linear phase modulation) is effectively a 1D translation model, indeed apparent in Fig 2. This is insufficient to capture 3D motion in practice.

The structure of the paper can be improved. Relevant information is now in appendices, and overhead in the main paper. For me, it is difficult to follow the main line of the paper.

The noise level used in experiments of sigma=0.4 is too high to arrive at meaningful results without smoothing out anatomical details compared to the ground truth.

In Fig. 3., VORTEX (Physics) appears to treat motion as noise, overly smoothing the image, including clinically relevant anatomical detail, e.g. in the meniscal area.



**Deanonymize Review:**

no

**Detailed Comments:**

Fig 2: it is unclear what the 'Image' represents, what type of transformation is applied? This is not explained in legend nor main text.

**Final Rating After The Rebuttal:**

5: Strong Accept

**Justification Of The Final Rating:**

I thank the authors for extending their experiments with an additional noise level, and consulting a radiologist to rate the scans given the measurement conditions. Certain other aspects can indeed be part of future work.

**Paper Type:**

methodological development

**Questions To Address In The Rebuttal:**

Can the authors comment on the limited applicability of the current motion model?

Please rerun experiments at a lower and more realistic noise level.

How does ‘motion’ propagate to sensitivity maps? For large translations, the sensitivity profiles for the data change significantly. Is this properly modelled/augmented?

**Special Issue:**

no

---

### Official Review · Reviewer_CDUA · 2022-01-24

**Confidence:** 4
**Preliminary Rating:** 4
**Recommendation:** Poster

**Summary:**

The paper proposes a physics-based MRI data augmentation framework named VORTEX that combines equivariant and invariant MRI perturbations (e.g., noise and motion image degradation) in order to increase performance of DL image reconstruction models.

As such, the authors demonstrate that their presented semi-supervised consistency training framework improves image reconstruction results when compared to conventional fully supervised techniques, whilst offering higher data-efficiency and robustness. Also, jointly integrating image-based and physics-driven data augmentation into model training, is shown to be superior to other recently proposed semi-supervised approaches.


**Strengths:**

The paper presents a solid methodology it is well written and clearly structured. The underlying MRI theory in terms of the encoding operator is nicely explained and formulated.

It is well described how the equivariant and invariant augmentation is combined in the training framework and how it is reflected in the loss functions.

Also, the paper demonstrates convincingly how both the physics-driven augmentations and the image-based augmentations can contribute to an improved reconstruction performance (including an ablation of the individual features of the VORTEX framework).
The authors present a comprehensive performance analysis of the proposed VORTEX scheme, including qualitative and quantitative comparisons with state-of-the-art DL reference methods, such as Noise2Recon, MRAugment or SSDU baselines.

Source code is available on github.


**Weaknesses:**

The results look promising. Nevertheless, they would be even more convincing if not only based on synthetic scenarios. In fact, I’d be curious to see how the model performs on real motion data, on prospectively undersampled datasets or how it generalizes to patient data with pathologies.

The authors state that 2D Poisson-Disc (under)sampling was used for model training and testing. Can you please motivate this choice in more detail? In particular, I am wondering how this type of undersampling relates to real prospective (coherent/incoherent) undersampling techniques.

Although the authors have conducted a thorough comparison with other DL-based methods, I am wondering how they compare to non-DL approaches, e.g. CS-based reconstructions.

Limitations of their work are not discussed at all.


**Deanonymize Review:**

no

**Detailed Comments:**

Figures 2 and 3 are very small.

Also, error maps would help to see the differences between the various methods better in Fig. 3.

I am missing a figure, which visually shows exemplary results for the FastMRI dataset (e.g. similar to Fig. 3 for the knee data).

The appendix gives VERY DETAILED insights into the competing DL-based image reconstruction methods. I am wondering if this level of detail is crucial. Given the comprehensive introduction and related works sections in the main part of the paper, a less detailed appendix might even improve the overall quality of the paper as more emphasis is put on its key aspects.


**Final Rating After The Rebuttal:**

4: Weak Accept

**Justification Of The Final Rating:**

With their work, the authors present a solid contribution to the field of MR image reconstruction. The methodology is sound and well presented. Also, the authors have addressed my remaining questions in their reply and the revised manuscript. Experimental results on synthetic data are promising. However, validation on real clinical data would be important to substantiate the significance of the work. That's why I am curious to see follow up work.

**Paper Type:**

methodological development

**Questions To Address In The Rebuttal:**

To further improve the quality of their paper, I would ask the authors to consider and discuss the aspects raised under weaknesses and detailed comments during the rebuttal period and to reflect on the limitations of their work.



**Special Issue:**

no

---

### Official Review · Reviewer_FtuM · 2022-01-25

**Confidence:** 4
**Preliminary Rating:** 5
**Recommendation:** Best Paper Award, Oral

**Summary:**

In the paper the authors present a new type of deep learning architecture for reconstructing undersampled/corrupted MRI data. The novelty in the presented work is that they propose applying physics-driven data augmentations during training based on domain knowledge of the forward MRI data acquisition process and to model relevant distribution shifts better. The proposed reconstruction framework is evaluated on a publicly-available mridata 3D fast-spin-echo (FSE) multi-coil knee dataset as well as the 2D fastMRI multi-coil brain dataset. The algorithm improves upon other standard methods and especially seems to improve data-efficiency and robustness to clinically relevant distribution drifts.

**Strengths:**

The paper highlights a very relevant problem in MRI acquisition where corrupted raw image data needs to be improved upon in order to reach diagnostic image quality.  The strengths to highlight are:

1) While many DL MRI reconstruction methods are relying solely on image-based data augmentations, it is refreshing to read about this approach to include physics-driven data augmentations as well. It is also nice to see in the results that this as expected improves upon the purely image-based approaches.

2) The paper is well written and covers all the necessary basics, especially the literature review is thorough while still leaving enough place for the methods and the experimental description.

3) It is refreshing to actually see that the authors are providing the code and behind the link is a reasonable github repository. Sharing the trained models for several different scanner settings would be the next step forward!


**Weaknesses:**

There are only a few weaknesses:

1) In the results presented in table 2 only the mean SSIM and PSNR values are given for the first dataset, but wy are the standard deviations or standard errors not included? And furthermore, since this is all paired data you could easily visualize this table in box plots and also at least for some of the improvements do statistical testing with a non-parametric paired test to see if you have a significant improvement upon e.g. the baseline image-based approaches. Although of course multiple comparisons is an issue when trying to do this for all.

2) Noise and motion levels were chosen "based on visual inspections of clinical scans". Can this be elaborated? You used different augmentations with different values on the training images and then visually compared it to clinical scans?

3) Your current training data consists of fully-sampled examples in k-space y(s) with corresponding supervised reference ground truth images x(s), and undersampled-only k-space examples y(u). Now through your augmentation you are trying to turn the undersampled-only k-space examples to more appropriate real examples. But what about additionally including clinically relevant, real corrupted data? Either form the clinic or from publicly accessible challenges such as https://realnoisemri.grand-challenge.org/?

4) Regarding the motion modeling, it seesm you are only modeling translation in image, aka shifts in k-space. Bt what about including, rotations in image, so phase ramps in k-space? Rotations are almost always part of the problem when e.g. imaging children (see e.g. https://www.frontiersin.org/articles/10.3389/fradi.2021.789632/full).


**Deanonymize Review:**

yes

**Final Rating After The Rebuttal:**

5: Strong Accept

**Justification Of The Final Rating:**

I thank the authors for their thorough responses to me and my fellow reviewers and still agree with my original assessment. I also like one of the others would like to see this used on clinical data, but also think this work is very important in itself. And I also agree with the issue raised by the authors that currently clinical data for testing their method on is limited. And after checking it is true that the realnoisemri challenge data is embargoed and hence could not be used.

**Paper Type:**

methodological development

**Questions To Address In The Rebuttal:**

I would like the authors to address the few weaknesses mentioned above, especially the presentation of the results and their statistical validity. Furthermore, I would like them to elaborate on my points 2-3.

**Special Issue:**

yes

---

### Meta-Review · Area_Chair_2yD6 · 2022-02-13

**Recommendation:** Accept (Oral)
**Confidence:** 5

**Metareview:**

I would like to thank all reviewers for their time and effort spent reviewing the paper and for their engagement with the rebuttal process. I also thank the authors for their detailed rebuttal and the changes made to the manuscript. All three reviewers agree that the paper should be accepted so I am happy to go with their recommendation and look forward to seeing the paper presented at MIDL.

---

### Decision · Program_Chairs · 2022-02-28

Accept